

# Targeted metabolomics suggests a probable role of the FTO gene in the kynurenine pathway in prediabetes

La-or Chailurkit,  Suwannee Chanprasertyothin,  Nisakron Thongmung,
Piyamitr Sritara and  Boonsong Ongphiphadhanakul

Ramathibodi Hospital, Bangkok, Thailand

## ABSTRACT

**Background**. Genome-wide association studies have identified the alpha-ketoglutarate dependent dioxygenase gene (FTO) as the first susceptibility gene of obesity. In the present study, we utilized targeted metabolomics in an attempt to further elucidate mechanisms underlying the action of the FTO gene.

**Methods**. This study was part of a health survey of employees of the Electricity Generating Authority of Thailand ($n = 79$, 10 female and 69 male). Targeted metabolomics was performed by liquid chromatography–mass spectrometry using Biocrates AbsoluteIDQ-p180 kit. Genotyping of FTO rs9939609 was performed by real-time PCR (TaqMan[TM] MGB probes).

**Results**. Using OPLS-DA variable importance in projection (VIP), tryptophan was found to be among the metabolites with the 10 highest VIP scores. Pearson's correlation analysis showed that kynurenine and tryptophan were positively correlated only in subjects with the rs9939609 A allele ($n = 32$, $r = 0.56$, $p < 0.001$) and the correlation coefficients were significantly higher in subjects having the A allele than in those without the A allele ($p < 0.05$). Moreover, the kynurenine/tryptophan ratio was significantly associated with the presence of the A allele, independently of body mass index and sex.

**Conclusions**. The FTO gene is likely to influences the conversion of tryptophan to kynurenine.

# INTRODUCTION

Genome-wide association studies have identified the alpha-ketoglutarate dependent dioxygenase gene (FTO) as the first susceptibility gene of obesity (*Loos & Yeo, 2014*; *Dina et al., 2007*). However, the mechanisms leading from the body mass index (BMI)-associated FTO gene variants to adiposity are unclear. Previous studies suggested the role of FTO genetic variants in switching thermogenesis in adipose tissue (*Claussnitzer et al., 2015*). Moreover, the nuclear RNA N6-methyladenosine (m6A) demethylation by FTO may also be partly responsible for the effect of FTO on obesity (*Jia et al., 2011*). Previous studies suggested a role of FTO in nucleic acid repair enzyme (*Xiang et al., 2017*) and/or, RNA modification (*Berulava et al., 2013*), but how this leads to altered energy homeostasis is unclear. Energy homeostasis is regulated and/or, reflected in a number of circulating

Corresponding author
La-or Chailurkit,
laor.cha@mahidol.ac.th

metabolites. In experimental animals, branched-chain amino acids and tryptophan have been shown to influence energy metabolism (*Simonson, Boirie & Guillet, 2020*). In addition, free fatty acid levels are known to be increased in obesity and can cause insulin resistance in major insulin target organs (*Boden, 2011*). In recent years, there have been increasing efforts to discover biomarkers and mechanistic pathways of obesity using the simultaneous measurements of a large number of metabolites (*Park, Sadanala & Kim, 2015*). However, few studies have investigated the profile of metabolites based on the genetic variation of FTO (*Kim et al., 2016*). Metabolomic approaches have been used to examine the response to standardized metabolic challenges of metabolic fluxes (*Volkova et al., 2020*). However, only minor effects of the FTO gene in this regard was found (*Wahl et al., 2013*).

Obesity is a proinflammatory state (*Herron, 2005*) and circulating C-reactive protein is elevated in obesity. The role of the FTO gene on inflammation, or vice versa, is still not entirely clear. No association between genetic variations in the FTO and inflammatory markers was found (*Olza et al., 2013*). Inhibition of the FTO gene, however, can inhibit macrophage activation, inhibit NLRP3 inflammasome in a mouse model (*Luo et al., 2021*). Tryptophan can influence energy homeostasis and obesity (*Kałuzna-Czaplińska et al., 2019*) and the tryptophan kynurenine pathway is also related to inflammation. The relationship between the FTO gene and the tryptophan kynurenine pathway has been less explored. To further investigate the functions/roles of the FTO gene and their related pathways, we examined in the present study the metabolite profiles according to the FTO genotype, based on the rs9939609 SNP, which have been clearly shown in a number of previous studies to be associated with obesity (*Ali, Shkurat & Abbas, 2021*; *Tanofsky-Kraff et al., 2009*), using a targeted metabolomic approach.

## MATERIALS AND METHODS

### Subjects

A total of 79 subjects (69 males and 10 females) aged 25–77 years participated in this retrospective cross-sectional study. The subjects were randomly selected from the population studied in the Electricity Generating Authority of Thailand (EGAT) Study in 2009, cohort 3 (EGAT 3). Details of the study cohort have been elaborated and described previously (*Vathesatogkit et al., 2012*). Prediabetes was identified when subjects' fasting plasma glucose levels were higher than 100 mg/dL. Body mass index (BMI) was calculated as body mass (kg)/height (m)$^2$. Venous blood samples were drawn from each subject in fasting conditions. For serum samples, each sample was collected in a clean polypropylene tube and then centrifuged. Each sample was then aliquoted into a storage tube and left to stand at −80 °C until analysis. For buffy coat preparation, blood samples were drawn into EDTA tubes and then centrifuged. After removal of the plasma, the upper cell layer (buffy coat) was aliquoted into storage tubes and kept at −80 °C until use. This study was approved by the Committee on Human Rights Related to Research Involving Human Subjects, Faculty of Medicine, Ramathibodi Hospital, Mahidol University (COA. No. MURA2019/267) and it conformed to the provisions of the Declaration of Helsinki (as revised in Fortaleza, Brazil, October 2013). All experiments were performed in accordance

with relevant guidelines and regulations. All participants gave their written informed consent before participating in the study.

## High-sensitivity cardiac C-reactive protein (hsCRP)

Serum hsCRP was determined by a high-sensitivity latex-enhanced immunonephelometric assay (BN 100 Nephelometer; Dade Behring Inc., Derfield, IL, USA) with a detection limit of 0.1 mg/L, intra-assay coefficient of variation of $< 4.5\%$, and inter-assay coefficient of variation of $< 5.8\%$.

## FTO genetic analysis

DNA was extracted from the buffy coat specimens by the phenol–chloroform method. Individual genotyping of all subjects was performed using real-time PCR (TaqMan[TM] MGB probes; Applied Biosystems, Foster City, CA, USA). Ten nanograms of DNA was added into the PCR reaction, consisting of TaqMan Universal Master Mix ($1\times$) and TaqMan MGB probes for FTO rs9939609 SNP ($1\times$) in a total volume of 10 µl. The real-time PCR reaction protocol was 10 min at 95 °C, 40 cycles of 15 s at 92 °C, and 1 min at 60 °C using a 7500 Real-Time PCR System (Applied Biosystems, Foster City, CA, USA).

## Targeted metabolite assessment (as previously described *Chailurkit et al., 2020*)

Subjects were grouped according to the presence or absence of the FTO rs9939609 A allele. Serum targeted metabolomics was investigated using Absolute*IDQ*[TM] p180 kit (Biocrates Life Sciences AG, Innsbruck, Austria) by liquid chromatographic (LC) separation prior to tandem mass spectrometry measurements (MS/MS). The sample preparation and measurements were performed in accordance with the manufacturer's manual of the p180 kit. Liquid chromatography was performed on an Agilent Zobax Eclipse XDB C18, 3.0 mm × 100 mm 3.5 µm column fitted with an Phenomenex, C18, 4.0 mm × 3.0 SecurityGuard guard column. Mass spectrometry analyses were performed on a QTRAP 5500 (ABsciex, Framingham, MA, USA) using electrospray ionization operating in multiple reaction monitoring (MRM) mode coupled to an Agilent 1260 Series HPLC (Agilent Technologies, Santa Clara, CA, USA) controlled by Analyst 1.6.2 software (AB Sciex, Framingham, MA, USA). The amino acids and biogenic amines were measured by LC-MS/MS in positive mode, while the lipids, acylcarnitines, and hexoses were analyzed by flow injection analysis tandem mass spectrometry (FIA-MS/MS) in both positive and negative modes. Metabolites were quantified in accordance with the manufacturer's protocol using Met*IDQ*[TM] Carbon software for targeted metabolomic data processing and management. The assay quantifies up to 188 targeted metabolites covering the following six compound classes, including 21 amino acids, 21 biogenic amines, 40 acylcarnitines, 90 glycerophospholipids, 15 sphingolipids, and one hexose. Lipid side-chain composition is given in the format $Cx{:}y$, where $x$ denotes the number of carbons in the side chain and y the number of double bonds.

## Statistical analysis

Statistical analysis was performed and predictive models were constructed using RStudio version 1.0.136 and R version 3.3.2 (*R Core Team, 2016*; *RStudio Team, 2017*). Comparisons

**Table 1** Clinical characteristics of the study population ($n = 79$).

| Variable | Mean ± SD or number (%) |
|---|---|
| Age (year) | $41.0 \pm 6.8$ |
| Male (%) | 69 (87.3%) |
| Body mass index (kg/m²) | $26.9 \pm 4.4$ |

between groups were performed by Student's t test. Feature selection was performed using a partial least squares-based algorithm for parsimonious variable selection, with variable importance in projection (VIP). Orthogonal partial least-squares regression discriminant analysis (OPLS-DA) modeling of baseline metabolites for the presence or absence of the FTO rs9939609 A allele was performed with the ropls R package. The significance of R2 and Q2 values was estimated by permutation testing. The features selected were then analyzed for statistical significance using univariate logistic regression or linear regression analyses, as appropriate. $P < 0.05$ was considered to represent a statistically significant difference.

# RESULTS

Clinical characteristics of the study population are shown in Table 1. Most of the subjects in the study population were male because of the employee demographics of the Electricity Generating Authority of Thailand. BMI tended to progressively increase with the number of the FTO rs9939609 A allele, but this did not reach statistical significance ($P = 0.13$), as shown in Fig. 1.

We then used OPLS-DA to examine the distribution of metabolites with regard to the FTO genotype. Table 2 shows the variable importance in projection (VIP) score according to absolute scoring of the studied metabolites and the FTO genotype from the OPLS-DA analysis. The score plot of the analysis is shown in Fig. 2. As tryptophan was among the metabolites within the three highest VIP scores, we further examined the influence of the FTO gene on the major pathway of tryptophan catabolism, the kynurenine pathway.

Using multiple logistic regression analysis, it was found that both lysophosphatidylcholine-acyl C16:1 and tryptophan were independently related, albeit in opposite directions, with the presence of the FTO A allele after controlling for age, BMI, and sex. No independent effect of ADMA was found (Table 3), while no association between lysophosphatidylcholine-acyl C16:1 ($R2 = -0.01$, $P = 0.99$) was found and the association of tryptophan with BMI just reached statistical significance ($R2 = 0.04$, $P = 0.05$).

As the kynurenine pathway is the main metabolic pathway of tryptophan, we further investigated the association of the FTO genotype with this pathway. Pearson's correlation analysis showed that kynurenine and tryptophan were positively correlated only in subjects with the rs9939609 A allele ($n = 32$, $r = 0.56$, $P < 0.001$), but not in those without it ($n = 46$, $r = 0.20$, $P = 0.20$). The correlation coefficient was significantly higher in subjects having the A allele than in those without it ($P < 0.05$).

Systemic inflammation is known to affect the tryptophan-metabolizing enzyme indoleamine 2,3 dioxygenase, which plays a key role in the conversion of tryptophan to kynurenine. Systemic inflammation, assessed using circulating hsCRP in the present

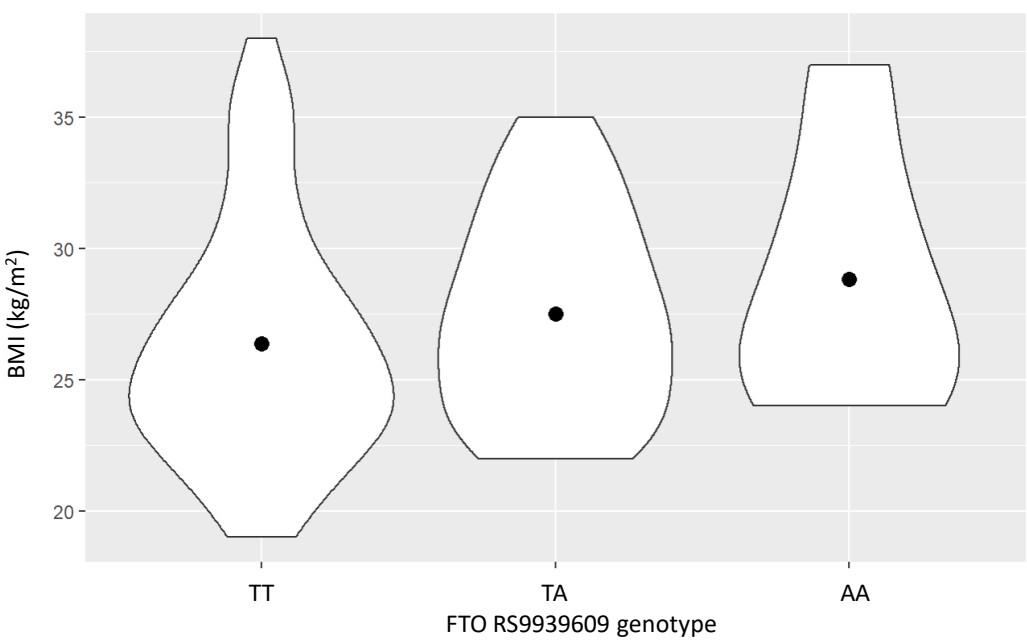

**Figure 1** The relationship between the FTO rs9939609 genotype and body mass index (BMI) (mean ± SE).

**Table 2** Three metabolites with the highest variable importance in projection (VIP) scores from OPLS-DA.

| Metabolite | VIP score |
| --- | --- |
| Lysophosphatidylcholine-acyl C16:1 | 2.67 |
| Asymmetric dimethylarginine | 2.41 |
| Tryptophan | 2.31 |

study, was significantly associated with the kynurenine-to-tryptophan ratio, an index of the conversion from tryptophan to kynurenine ($r = 0.33$, $P < 0.01$). The FTO A allele was still significantly associated with the kynurenine-to-tryptophan ratio, even after adjusting for age, BMI, sex, and serum hsCRP (Table 4).

## DISCUSSION

Tryptophan exerts many biological effects, including those related to obesity and energy homeostasis (*Kałużna-Czaplińska et al., 2019*). We found in the present study that genetic variation in the FTO gene is likely to influence the kynurenine pathway, the main metabolic pathway of tryptophan. This finding is in keeping with a number of lines of evidence (*Badawy, 2017*; *Muneer, 2020*; *Gostner et al., 2020*). For example, a diversion of kynurenine toward kynurenine monooxygenase activation by adipose tissue macrophages has been demonstrated in obesity (*Favennec et al., 2015*). Tryptophan is converted to kynurenine by the action of the enzymes tryptophan 2,3-dioxygenase (TDO) and indoleamine 2,3-dioxygenase (IDO). As the kynurenine/tryptophan ratio differed

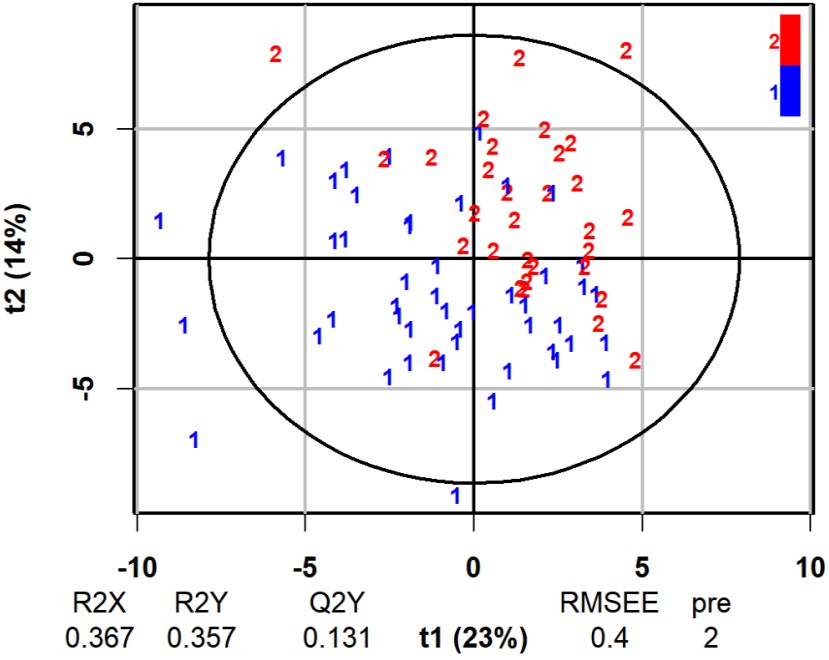

1 = FTO rs9939609 A allele -ve
2 = FTO rs9939609 A allele +ve

**Figure 2** Score plot of the analysis of the FTO A allele classification based on serum metabolites.
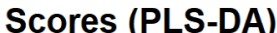

**Table 3** Independent associations of lysophosphatidylcholine-acyl C16:1 and tryptophan with the presence of the T allele in the FTO gene from multiple logistic regression analysis.

| Variable | Coefficient | 95% CI | p value |
|---|---|---|---|
| Age (year) | $-0.07 \pm 0.05$ | $-0.16$ to $0.02$ | 0.14 |
| Body mass index (kg/m$^2$) | $-0.02 \pm 0.08$ | $-0.17$ to $-0.13$ | 0.76 |
| Female | $0.99 \pm 0.91$ | $-0.79$ to $2.77$ | 0.26 |
| Lysophosphatidylcholine-acyl C16:1 | $-0.002 \pm 0.001$ | $-0.004$ to $-0.001$ | $<0.01$ |
| Asymmetric dimethylarginine | $-0.016 \pm 0.011$ | $-0.038$ to $-0.005$ | 0.141 |
| Tryptophan | $0.0006 \pm 0.0002$ | $0.0003$ to $0.0009$ | $<0.001$ |

between the two FTO alleles in the present study, it is likely that these enzymes could mediate the role of the FTO gene in tryptophan metabolism.

In regard to systemic inflammation, obesity is a proinflammatory state and the IDO-1 enzyme is known to be activated by inflammation. Moreover, systemic inflammation has been shown to induce a number of changes in circulating metabolites and their associated pathways. In line with this, in the present study we found that systemic inflammation, as reflected by serum hsCRP, was highly correlated with the kynurenine/tryptophan ratio. It is also possible that the interaction between the FTO genotype and the conversion

**Table 4  Associations of tryptophan and high-sensitivity cardiac C-reactive protein (hsCRP) with the kynurenine/tryptophan ratio from multiple linear regression analysis.**

| Variable | Beta coefficient | $p$ value |
|---|---|---|
| Age (year) | 0.15 | 0.18 |
| Body mass index (kg/m$^2$) | 0.13 | 0.27 |
| Female | −0.07 | 0.53 |
| lnhsCRP | 0.36 | <0.01 |
| FTO allele | −0.23 | <0.05 |
| Tryptophan | 0.0006 ± 0.0002 | <0.001 |

of tryptophan to kynurenine, as assessed by the kynurenine/tryptophan ratio, could be mediated through systemic inflammation. Nevertheless, this is refuted to some extent as the association of the FTO genotype with the tryptophan-to-kynurenine conversion was still statistically significant after controlling for serum hsCRP.

Phospholipids are associated with obesity and lysophosphatidylcholine-acyl C16:1 as well as lysophosphatidylcholine-acyl C16:0 have been demonstrated to be positively related to brown adipose tissue volume and activity (*Boon et al., 2017*). In addition, the literature describes a decrease of lysophosphatidylcholine species in plasma of obese subjects and a potentially anti-inflammatory role of lysophosphatidylcholine in these subjects (*Heimerl et al., 2014*). Our results are in keeping with these previous findings in that we found that lysophosphatidylcholine tended to decrease with BMI. It is unclear whether the association between the FTO genotype and lysophosphatidylcholine was the result of the association of FTO with obesity. Nevertheless, our findings contribute to the ongoing discussion about the role of lysophosphatidylcholine in obesity and related chronic inflammation, strongly supporting pre-existing data in the literature that show a decrease of lysophosphatidylcholine species in plasma of the obese and a potentially anti-inflammatory role in these subjects (*Wahl et al., 2013*). However, in the present study, lysophosphatidylcholine C16:1 was not associated with BMI, but was associated with the FTO alleles, so further investigation is warranted.

There are a number of limitations in the study. The sample size was small as the study was intended to be an initial exploratory study. Moreover, it cannot be determined if the relationship found between the FTO gene and the kynurenine pathway has any causal or reverse causal relationship. It is still possible that the relationship found could be just a correlation due to other underlying confounders. Further experimental studies are therefore warranted.

## CONCLUSION

We conclude that the FTO gene may influence the conversion of tryptophan to kynurenine. Alteration of the kynurenine pathway may be one of the mechanisms underlying the action of FTO in the pathogenesis of obesity and its related effects.

### Funding

This study was supported by the Mahidol University Research Grant. The funders had no role in study design, data collection and analysis, decision to publish, or preparation of the manuscript.

### Grant Disclosures

The following grant information was disclosed by the authors:
Mahidol University Research Grant.

### Competing Interests

The authors declare there are no competing interests.

### Author Contributions

- La-or Chailurkit performed the experiments, analyzed the data, prepared figures and/or tables, authored or reviewed drafts of the article, and approved the final draft.
- Suwannee Chanprasertyothin performed the experiments, authored or reviewed drafts of the article, and approved the final draft.
- Nisakron Thongmung performed the experiments, authored or reviewed drafts of the article, and approved the final draft.
- Piyamitr Sritara performed the experiments, authored or reviewed drafts of the article, and approved the final draft.
- Boonsong Ongphiphadhanakul conceived and designed the experiments, analyzed the data, prepared figures and/or tables, authored or reviewed drafts of the article, and approved the final draft.

### Data Availability

The raw data is available in the Supplemental Files.

### Supplemental Information

Supplemental information for this article can be found online at http://dx.doi.org/10.7717/peerj.13612#supplemental-information.

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
