# Peer review of "Targeted metabolomics suggests a probable role of the FTO gene in the kynurenine pathway in prediabetes"

_PeerJ, doi:10.7717/peerj.13612_

## Round 0.1 · original submission · Major Revisions

The reviewers have evidenced several issues that need to be addressed, especially in the methodology and interpretation of results. Limitations of the study need to be discussed, including bias in the experimental design, such as selection of patients.

·

Basic reporting

- I think this manuscript is easy to follow. However, some clauses in the Introduction should cite references. For example, in lines 39-40, you wrote "Genome-wide association studies have identified the fat mass and obesity-associated gene (FTO) as the first susceptibility gene of obesity. ". Then, lines 52-53 "However, 53 few studies have investigated the profile of metabolites based on the genetic variation of FTO. " also deserve references. Please add them.

- From the Introduction, I understood the rationale for analyzing the FTO gene. However, how did you select this FTO rs9939609 SNP?

Experimental design

- In the Subjects section, you described only a serum collection. However, DNA extraction for genotyping purposes is usually performed with buffy coat samples. Did you collect buffy coats from these patients? It should be added to the Subjects section.

Validity of the findings

- In lines 124-125, you stated "BMI tended to progressively increase with the number of the FTO
rs9939609 A allele, but this did not reach statistical significance, as shown in Figure 1". Which was the p-value? Have you tested other genotypic models (allelic, dominant, and/or recessive)? By the way, I looked at your raw data. I did not find BMI values. Could you add them, please?

Additional comments

- I think you have enough results for further analysis. I would suggest testing regression models involving all significative variants aiming to build a fast tool (such as a nomogram). In my view, it would take advantage of your results.

Reviewer 2 ·

Basic reporting

Line 39. Needs a reference. Databases like the GWAS catalog website could also be used.
Line 20: The most updated name for Fat mass and obesity-associated gene FTO is alpha-ketoglutarate dependent dioxygenase gene.
Line 50-52, 53, 55, 155, also need references to according to studies.

No information or details regarding the specific genetic variation in the FTO gene is provided in introduction. For example, why among all the SNPs in the FTO gene this one in particular is chosen.
Moreover, no details about systemic inflammation which is in the next sections are one of the focuses of this study is provided in the introduction.

Some sentences as noted below have complicated phrasing.
I suggest better English proofreading for these lines 43, 44, 45, 48,49, 131, 154-7, 162-4, 167-70, 180-3.
Line 44. the word "role" instead of "action" would better describe the meaning. Also "and or comma" instead of "or" in lines 45 and 47.

Line 52. To reduce the ambiguity of sentences, it should be stated the phrase "physiological pathways" underscores which physiologic eras (Obesity?).

Line 65. Any details for the study cohort are directed to a previously described article in reference No.8. It is most acceptable to describe the significant aspects of the study cohort through the method section.

Also, the referenced paper for the population cohort, (Ref No. 8) presents the "EGAT 1" population, which is irrelevant to this study and is published in 2007.
The population studied here is "EGAT 3" which started in 2009.

There is a lack of description of variables in methods section. "Progression to diabetes 3 years later" first is seen on line 67 which its contribution in the study has not been articulated before.

There is no straightforward list of metabolites that are analyzed. They could only be found as column headers in supplementary materials. This list is best to be more accessible for one who is reading the article.

The diabetic definition is not consistent, progression to diabetes is first described as progression in 3 years, but in the statistical section 5 years is mentioned.

In the description of each table, it is better to include the name of the analysis of the statistical test in which it is done for those data and in Table 3 there is a typing error before 0.13.

Line 147. FTO G is misspelled and is not clear which of A or T alleles are meant to be here.

Line 173. The word "Phospholipid" is not clear to which of phospholipids is pointing to.

It is unclear why "Prediabetes" and "Progression to diabetes" were defined in the method section since these entities were not included in the downstream analysis in the results and discussion sections. Providing further information about inclusion/exclusion criteria would address this issue.

Experimental design

Line 56. The description of the study's hypothesis should be more clarified. Also, "Functions" could be changed to "function"/"role".
The grouping of participants is not clear and should be indeed described in the method section.

It is not clear why the study focus is redirected to tryptophan after OPLS-DA analysis. Although the FTO gene pathway, the Kynurenine pathway, is associated with tryptophan and serotonin metabolism, the reasoning behind this approach of focus on Tryptophan is not well described through the study background.

Validity of the findings

Line 122. As in table 1. Most subjects in the study are female, not male. Although this info is inconsistent through the submission. In Table.1 males are 68, in Line 62 males are 9, and in abstract males are 70.

No information regarding the VIP scores for other metabolites is provided.
It is not clear if these three top metabolites are the top three sorted for highest in positives or highest in absolute scoring.

Furthermore, score plots of variable selection analyses are so informative which are missed here.

Table 3. The highest association coefficient from logistic regression is for Gender (0.99), which is 100 times bigger than the highest among metabolites. Considering the imbalance of the female/male ratio (1:8) of study participants it is best to subdivide the population among males and repeat the logistic regression only in the male group. This will increase the power for detecting the lowest association coefficients such as the observed numbers for metabolites here.

Line 141-142, For comparing the Pearson's correlation values, we need to transform the values in Fisher’s r to z transformation. Nor in the method statistical section nor in the results section it is mentioned if this transformation is applied here or not. Although even if this method had been applied, while one of the correlation p-values is not significant, any conclusion from this transformation is still limited.

Line 147. Although the P-value of the correlation is below 0.05. But r is 0.03 which is inconsistent with the presented significant association between CRP and metabolites ratio.

Line 177,178. Among the presented data and tests in this manuscript, no findings suggest the conclusion in lines 178, 186.

Additional comments

The limitations of the study are not implied and no recommendations for further investigations are provided. The potential confounder effect of fasting before blood sampling is not articulated.
Furthermore, Tryptophan itself, and the Kyrunilic pathway are associated with serotonin production. Also, Quinolinic acid, one end production of the Kyrunilic pathway could be tightly associated with psychiatric, mood, and eating disorders. These associations could cause bias if not considered in the study design of research on obesity.

---

## Round 0.2 · Minor Revisions

The authors have satisfactorily addressed most concerns raised by the reviewers. However, a few points from reviewer 1 still need to be clarified.

·

Basic reporting

Dear authors,

Thank you for addressing my first concerns. However, I would like to comment on the following points:

- In the "Subjects" section, methods for obtaining buffy coats from patients should be added.

Experimental design

- Despite having few samples to produce a nomogram, I think it is necessary to do a multivariate analysis on Tables 3/4. Please consider applying this test to avoid overinterpretation.

Validity of the findings

- Thank you for clarifying interpretations of p-values and sending raw data. In addition, I think that mean+- the standard error could hide some data. For figure 1, I suggest changing this plot for a violin plot. It would improve the communication of your findings in a more transparent form.

Reviewer 2 ·

Basic reporting

Authors did an excellent job improving the professional article structure and in their response to my previous comments, they have clarified the manuscript ambiguities and the issues in the context of clarity, data reproducibility are now addressed.

Experimental design

No Comment

Validity of the findings

No comments

---

## Round 0.3 · accepted · Accept

The authors have satisfactorily clarified the last points left.

·

Basic reporting

Dear authors,

Thank you for having addressed my last comments. In their current version, I endorse this manuscript for its publication.

Experimental design

No comment

Validity of the findings

No comment